# Evaluation of Biodegradable PVA-Based 3D Printed Carriers during Dissolution

**DOI:** 10.3390/ma14061350

**Published:** 2021-03-11

**Authors:** Bálint Basa, Géza Jakab, Nikolett Kállai-Szabó, Bence Borbás, Viktor Fülöp, Emese Balogh, István Antal

**Affiliations:** Department of Pharmaceutics, Semmelweis University, Hőgyes E. Street 7-9, 1092 Budapest, Hungary; basa.balint@pharma.semmelweis-univ.hu (B.B.); jakab.geza@pharma.semmelweis-univ.hu (G.J.); kallai.nikolett@pharma.semmelweis-univ.hu (N.K.-S.); borbas.bence@stud.semmelweis-univ.hu (B.B.); fulop.viktor@pharma.semmelweis-univ.hu (V.F.); balogh.emese@pharma.semmelweis-univ.hu (E.B.)

**Keywords:** 3D printing, fused deposition modelling (FDM), computer aided design (CAD), erosion test, dissolution study, dynamic light scattering (DLS)

## Abstract

The presence of additive manufacturing, especially 3D printing, has the potential to revolutionize pharmaceutical manufacturing owing to the distinctive capabilities of personalized pharmaceutical manufacturing. This study’s aim was to examine the behavior of commonly used polyvinyl alcohol (PVA) under in vitro dissolution conditions. Polylactic acid (PLA) was also used as a comparator. The carriers were designed and fabricated using computer-aided design (CAD). After printing the containers, the behavior of PVA under in vitro simulated biorelevant conditions was monitored by gravimetry and dynamic light scattering (DLS) methods. The results show that in all the dissolution media PVA carriers were dissolved; the particle size was under 300 nm. However, the dissolution rate was different in various dissolution media. In addition to studying the PVA, as drug delivery carriers, the kinetics of drug release were investigated. These dissolution test results accompanied with UV spectrophotometry tracking indirectly determine the possibilities for modifying the output of quality by computer design.

## 1. Introduction

Among the challenges of pharmaceutical technology and drug development, the desire to produce the perfect treatment fit for the individuals via the construction of the drug delivery system is inevitable. The aspect of producing personalized medicine with optimal pharmacokinetics and physicochemical properties engineered strictly to the patient’s needs is gaining more and more attention. 3D printing can change the ways of traditional drug production. The precedent of Spritam^®^, the first 3D printed pharmaceutical product, shows the enormous potential which hides in so-called Additive Manufacturing (AM) [1]. During this process, the number of unit operations is minimalized [2], and the opportunity to fabricate every single prototype shaped according to the individuals’ profile with only minimal human intervention can be the cause of the increased research activity in this field [3]. An additional benefit of this type of manufacturing is the capability of producing customized ways of medication for patients suffering from organ dysfunctions, avoiding the slightest chance of reaching toxic doses in their body. Moreover, preferred patient groups are pediatrics [4] and geriatrics [5] where therapeutic doses perform great variance [6]. Further fields of interest can be the production of orphan drugs due to the low amount of produced medication. With the utilization of 3D printing, these single batches can be microfabricated without retooling all the manufacturing devices [7]. However, revolutionizing pharmaceutical manufacturing also requires a new regulatory attitude [8].

A vast number of methods are available beyond 3D Printing, and few of them which can be employed for tailored pharmaceutical manufacturing [9]. The most widely investigated type of free form fabrication is extrusion printing, in which the technique can be divided by the step of melting the used materials or not. The Pressure Assisted Microsyringe System (PAM) utilizes the components without melting them [10], while Fused Deposition Modeling (FDM) uses the melted excipients to create the layer-by-layer structure of the 3D object [9]. The mixture of active pharmaceutical ingredients (API), polymers, and other excipients gives a great opportunity to modify the viscosity of the preprinted materials, and therefore there is no need for melting the semi-solid substrates [11]. Via the formulation of the containers, several modified releasing strategies can be applied in the same printlet. The idea of combining osmotic release items into a diffusible containing shell gives a great opportunity to tailor the released doses according to the patient’s condition [12]. Moreover, the idea of producing “polypills” containing at least three or more different APIs with different release profiles also seems to be reachable with this technology [13]. In the case of harmonizing these strategies, a whole new dimension of pharmaceutical engineering and manufacturing can be established [2]. Another approach of this layering method is melting the thermoplastic polymers just above their melting temperature, then the melted excipients adhere to the heatable printing bed [14]. Due to the adaptability of the heat of the printing nozzle, the resolution of the printing is much better than in the instance of PAM. The pre-designed modifications inside the CAD file are easier to execute because of the better printing. The great variability of printable filaments and the low cost of this type of manufacturing generated an enormous breakthrough in the field of pharmaceutical manufacturing [15]. Among the most critical printing parameters, the infill and its patterns [16] and the height of the printed layers [17] can be an outstanding opportunity to modify the release kinetics of the microfabricated drugs [18]. In the development of patient-friendly drug delivery, designing and producing various geometries with a standard quality in order to maintain different kinetics is an important objective [19]. FDM printing gives the chance to fabricate unique designs for each object, and these delivery systems can be tailored to the individual’s preferences [20]. The FDM extrusion-based technology includes a heating step in which the metal extruder can reach high processing temperature, excluding the possibility of printing thermolabile API filled filaments [21]. In order to formulate dosage forms suited for heat-sensitive APIs, novel nanocomposites have been developed lately [22]. Thermostable APIs can be impregnated onto the surface of the filament, which allows a minimal drug loading percentage [23,24]. The other and widely used method is hot-melt extrusion, where the parallelly co-rotating extruders make the homogenous drug-loaded filaments ready for printing [25]. In most studies, this preparation step was the basis of fabricating tablets and capsules having different geometries containing variant drugs and doses [26,27,28]. 

Beside producing drug-loaded filaments, another way to place API into the delivery system is printing separate capsule or carrier parts then filled and assembled in the post-printing phase [29,30,31]. There was only one previous formulation which aimed at printing a capsule which can be filled with liquid or solid API and excipients mid-printing [32]. Since, in case of this study, the basis of the formulation strategy is a thin-walled carrier, the buckling behavior of the printed structures should be monitored in order to ensure the desired quality [33].

Several types were previously microfabricated: immediate-release tablets [34], fast-disintegrating tablets and orodispersible films [35], floating drug delivery systems [27,36], pulsatile drug release tablets [30], biphasic and multi-active solid dosage forms, and zero-order release tablets [37]. Overcoming the difficulties of on-demand manufacturing of personalized carriers can lead to the spread of the clinical application of pharmaceutical additive manufacturing [38]. The reproducibility of dose dispensing and carrier filling is promising for the future [39]. 

Polyvinyl alcohol (PVA) is a non-toxic, hydrophilic, synthetic, biodegradable polymer produced via the hydrolysis of vinyl acetate [40], which is the most widespread supporting material in the field of fused deposition modeling. The variants of PVA are usually chosen as supporting structure due to its good solubility in water. The spread of biodegradable excipients in the industrial production not only decreases the ecological footprint [41] but is also an adequate step towards a sustainable, zero-waste manufacturing [42]. Polyvinyl alcohol is used primarily in topical pharmaceutical and ophthalmic formulations [43]. It has also been used as an emulsifier in the formulation of drug loaded micro sponges [44]. In solid dosage forms PVA is used in coating formulations for tablets as a film forming polymer [45]. In this study our aim was to characterize the erosion of water-soluble PVA-based 3D printed systems with particle size analysis of colloidal PVA particles which appeared during in vitro mimicked dissolution conditions. The effect of pH, and the presence of bile salts were also simulated. An additional objective was to evaluate the effect of orifice numbers on the riboflavin release as a function of time. The optimal setting of this adjustment through CAD design ensures a perfect dose release which is inevitable for personalized therapy. The results of these investigations are intended to promote the spread of the 3D printed production of fillable water-soluble shells. With the opportunity of mixing different API between different layers of the carrier, the individualized medication can gain more emphasis.

## 2. Materials and Methods

Two commercially available biodegradable polymers were used as printing filament: the water-soluble polyvinyl alcohol (PVA) fiber (Orbi-Tech advanced, Leichlingen, Germany) and the water-insoluble polylactic acid (PLA) fiber (bq Easy Go, Madrid, Spain). The PVA filament has a diameter of 1.75 mm and a melting point of 183 °C, density: 1.13 g/cm^3^. According to the manufacturer’s specification, the orange PLA fiber had a diameter of 1.75 mm and a density of 1.24 g/cm^3^ (ASTM D792), while the melting point temperature is between 145 and 160 °C. As model API riboflavin (Hungaropharma, Budapest, Hungary) was used. Polyethylene glycol 300 (Sigma Aldrich, St. Louis, MO, USA) was applied as the dispersing agent of the API containing liquid filled into the 3D printed carriers.

### 2.1. Design

Autodesk Fusion 360 (Autodesk Inc., San Rafael, CA, USA) was used during the planning phase of the experiment which exported the prepared structure into a stereolithography file. Further settings should be applied during the slicing. This is an algorithmical step in which the slicing program Ultimaker Cura (Ultimaker, Geldermalsen, Holland) divides the designed object into several, well defined horizontal slices. The exported g-code file includes not only the coordinates the extruder is going to follow but all the other printing parameters which can be set, e.g., printing temperature, bed temperature, cooling fan speed, printing resolution, and the infill percentage.

The PVA and PLA-based carriers were designed as a 9.75 mm high by 9.6 mm in diameter. The structure of the wall was built by concentric movements making 1-mm-thick sides to the printed object. Each carrier was designed to have a hollow structure with approximately 0.4 mL reservoir volume. During the examination of the effects of holes, 1-mm-diameter orifices were designed into the cylinder mantle and top (Figure 1), equally divided around the circumference of the prototype. There was a version of carrier without orifices in order to investigate only the release-modifying effect of the wall. Carriers were also made of PVA and PLA containing 2 or more orifices (Figure 2). The distance between the orifices is the circumference of the cover circle divided by the number of holes. The printer process parameters were set as the following: print speed, 20 mm/s; travel speed, 120 mm/s. The layer parameters were set as a 0.16 mm layer height with a 0.2 mm layer thickness. The printing temperature was 200 °C, accompanied by a 50 °C bed temperature in order to ensure the perfect bonding to the plate. The infill percentage was set to 100%, with maximum cooling fan performance (5000 rpm).

### 2.2. FDM Printing

The applied 3D printer was Creality Ender 3 (Creality 3D Technology Co., Shenzhen, China), with an MK-10 hot-end with a 0.4-mm-diameter nozzle. The heated print platform was customized with 3M blue tape. The temperature during the printing was checked by FLIR with CAT S61 (Caterpillar, Deerfield, IL, USA) The frames were layered on each other with pre-selected points where the temperature was indicated. The visual investigations of the printlets were executed using a Keyence VHX-7000 digital microscope (Keyence International, Mechelen, Belgium).

### 2.3. Preparation of Riboflavin Containing Liquid Fill 

In every instance, 20 mg/g concentration of riboflavin-PEG 300 (polyoxyethylene glycol) dispersions were created manually in 10-g-sized batches. The samples were prepared before filling at room temperature with light-protection. The dispersion was applied into the vehicle using a 5 mL syringe (Chirana T. Injecta, Stará Turá, Slovakia) applied with a 23-gauge syringe needle. The filling was monitored by the comparison of the mass of the empty and the filled prototypes. In the instance of carriers without orifice, the filling process should be executed during the printing. The filament extrusion was stopped at 80% of the progress, and the dispersion was filled with the 5 mL syringe. After the application, the printing was continued, and the container was closed.

### 2.4. Physical Characterisation of 3D Printed Carriers

The 3D printlets are not yet subject to pharmacopoeial requirements; therefore, for the determination of mass uniformity and friability, the tests were performed as described in Chapter 2.9.5 (uniformity of mass of single-dose preparations) and Chapter 2.9.7 (friability of uncoated tablets) of the 9th European Pharmacopoeia for uncoated tablets, with an analytical balance (n = 20; Sartorius LA 230S, Sartorius AG, Göttingen, Germany) and tablet friability apparatus (Erweka AR, Langen, Germany). The printlets’ dimension (n = 20; Mitutoyo Absolute, Mitutoyo Corporation, Kawasaki, Japan,) and mechanical strength (n = 10; Erweka TBH 200 TD type, Erweka AR, Langen, Germany). The average values with standard deviation were recorded at each parameter test.

### 2.5. PVA-Based Carrier and Drug Release Study

The dissolution tests were performed using a Hanson SR-8 Plus™ Dissolution Test Station (Hanson Research, Los Angeles, CA, USA) with the paddle (USP 30 dissolution apparatus II) method at a rotation speed of 50 rpm, in 37 ± 0.2 °C medium of 500 mL volume. At predetermined time-points, 5 mL of samples were withdrawn and filtered through 10 µm pore size membrane full-flow filters from the media by Hanson^®^ AutoPlus Multifill collector (Hanson Research, Los Angeles, CA, USA). After every sampling, media replacement was accomplished with 5 mL of fresh buffer solution. The dissolution tests were performed in triplicates in the instance of every samples.

#### 2.5.1. Study of PVA-Based Carrier Erosion

The erosion and dissolution of empty PVA-based carriers were characterized by gravimetry, and to trace the number and particle size of PVA colloidal-sized aggregates dispersed in various dissolution media, the DLS (dynamic light scattering) method was utilized.

To evaluate the effect of ionic strength and surface-active agents on degradability, the erosion studies of the empty carriers were conducted in aqueous media of pure demineralized water, pH = 1.2 HCl, pH = 6.8 phosphate buffer and pH = 6.8 tris-(hydroxymethyl)-aminomethane (TRIS) buffer with or without sodium salt of bile acids (cholic acid and deoxycholic acid sodium salt 1:1 mixture (Sigma-Aldrich, St. Louis, MO, USA). The erosion of the carriers was tracked visually (Olympus Stylus TG-4 digital camera, Olympus Corp., Tokyo, Japan).

The erosion of the PVA-based carriers was tracked visually.

Filtered erosion samples were measured with the instrument Zetasizer Nano ZS™ (Malvern Instruments Ltd., Malvern, UK) for the derived count rate (DCR) and particle size. The Zetasizer^®^ instrument was equipped with a He-Ne laser (wavelength 633 nm, 4.0 mW) and an avalanche photodiode served as a detector at a detection angle of 173° (backscatter mode). Transmittance values for DLS were quantified by Agilent 8453 UV-Visible Spectrophotometer (Agilent Technologies Ltd., Santa Clara, CA, USA) at wavelength 633 nm. Measurement settings: automatic mode, NIBS (none-invasive-back-scattering) 173°, 30 sub runs/measurements; run duration: 10 secs, automatic laser position selected at 4.65 mm from the bottom of the cuvette; attenuation: attenuator 9 was selected automatically. Three measurements with 30 runs were performed for each sample, and the mean ± SD values are reported for all DLS parameters in this article. 

The weight of the 3D printed PVA carriers was determined on an analytical balance (Sartorius LA 230S, Sartorius AG, Göttingen, Germany), which will give the initial weight (*W_i_*) during the calculation. The carrier was then placed in the apparatus used in the in vitro dissolution test described above. The test was also performed under the conditions mentioned above. The dissolution medium was pH = 1.2. The printlets were taken out 5, 15, 30, 60, 120, and 240 min later; the dissolution medium was removed by vacuum filtration using a Pyrex^TM^ borosilicate glass filter. The residue was stored in an oven (6030 Heraeus Instruments GmbH, Hanau, Germany) at 70° C for 48 h. Dry printlet mass (*W_dry_*) was then determined (Sartorius LA 230S, Sartorius AG, Göttingen, Germany). The weight loss by erosion of carriers was calculated by Equation (1) respectively [46]:(1)$$weight\;\left(\%\right)=\frac{w_{dry}}{w_{i}}\times 100$$

*W_dry_*—mass of the dried printlet; *W_i_*—initial mass of the printlet.

#### 2.5.2. Riboflavin Release 

The riboflavin concentrations of the dissolution samples were measured by UV-spectroscopy (Agilent 8453 UV-Vis spectrophotometer; Agilent Technologies, Waldbronn, Germany) at 267 nm. 

Numerous theories and kinetic models describe and applied for the characterization of drug dissolution profile [47]. Since the shape of the investigated curves was different, the Weibull distribution function (1) was used for the characterization of the dissolution profile of the riboflavin loaded PVA or PLA-based 3D printlet [48].
(2)$$M_{t}=M_{\infty }\left[1-e^{-{\left(\frac{t-t_{0}}{\tau_{d}}\right)}^{\beta }}\right]$$

*M_t_*—the percentage of the dissolved API at time; $M_{\infty }$—the infinite concentration of the API in percentages; *t_0_*—dissolution lag time; *β*—curve shape parameter; *τ_d_*—time in minutes when 63.2% of the API has been dissolved.

Where *M_t_* is the percentage of the dissolved active pharmaceutical ingredient at time t, *M**_∞_* is the infinite concentration (%) of the drug, *t_0_* is the dissolution lag-time, *β* is the shape parameter of the curve, and the *τ_d_* represents the time (minutes) when 63.2% of the drug has been dissolved.

## 3. Results and Discussion

There are several formulations printed with polyvinyl alcohol, due to its soluble character [21,23,29,32]. However, there is extended research available in connection with the API–PVA formulations [49]; the erosion of PVA itself has not yet been described.

### 3.1. The CAD Design and the Tracking of the Printlet

The printing process (Figure 1) was captured by a FLIR thermal camera in order to obtain information about the already printed layers, while the upper ones are printed on the structure. The pinpoint set onto the carrier indicates that while significant heat comes from the nozzle and the heated bed towards the printlet, the state of the solidified layers is extremely acceptable. The temperature of the printed wall has not reached the 50 °C, and this phenomenon means that the filling of thermolabile API can be accomplished during the process.

### 3.2. Physical Characterisation of 3D Printed Carrier Systems

To guarantee the reproducibility of the manufacturing quality, the investigation of physical characteristics was executed on the printlets. As there is currently no official description in the pharmacopeia for the study of 3D printed carriers, we performed the study according to the pharmacopoeial description (Ph. Eur. 9.) of uncoated tablets.

The measurement results are shown in Table 1. For uniformity of mass, the standard deviation was minimal, well below the 5% allowed in the pharmacopeia. To check the print settings, it is also important to check the height and diameter of the 3D printed carrier. The measured values were close to the original value of the set parameters. The pharmacopoeia concedes a 1% weight loss for friability when testing uncoated tablets. Compared to this value, both PVA and PLA-based printlets had very low friability values, as shown in the data in Table 1. Besides replicability, these results indicate a good opportunity to produce fillable carriers not just for immediate usage but stock can be also piled from them. Due to the structure of the carriers, the weak point of this CAD design is the last layer of the wall around the hollow and the first layer of the closing top area of the printlet. The joint section of the two different layers ruptures if the hardness tests are executed. However, the lowest value of the hardness test performed was 205 N, while the highest was 350 N for PVA. The standard deviation surpasses the 5% limit; however, with the values oscillating in this territory, the mechanical behavior of the carriers shows no diverse differences. 

Digital microscopic images show the one-orifice PVA and PLA-based carrier (Figure 1). It is clear from the images that the layers formed according to the design file during FDM printing processes. The top view shows the designed orifice that plays an important part in the filling and dissolution of the active ingredient. Figure 2 shows the cross-section view of the biodegradable polymer carriers containing multiple drug delivery orifices design for 3D printing and the prepared prototypes.

### 3.3. Erosion of the PVA Carrier

As earlier mentioned PVA is commonly referred to as a water-soluble excipient, but PVA forms a physical hydrogel in an aqueous medium [50]. Of course, concentration conditions must be taken into account. Dilution of a physical hydrogel with water gives a colloidal solution. During the studies, our aim was to follow the behavior of the printed PVA carrier through the simulated circumstances of the GI tract. The third figure (Figure 3) shows what happens to a PVA-based carrier contacting aqueous medium. The colloidal dissolution/erosion of PVA is a consecutive process. The digital microscope image in the figure shows the wall of a PVA carrier located in a 90-minute release medium (pH = 1.2). Erosion and gel state of PVA can be observed. The solid-state wall is eroded into physical hydrogel state and very small fibers by the medium, forming a gel state before forming the colloidal solution. In the case of water-soluble PVA, the weight of the carrier decreases continuously (black line), while the DCR value determined from the release medium increases (red line, violet line). It is known in the literature that the increase in DCR is due to an increase in the concentration of the dispersed particles and/or an increase in the size of the dispersed particle [51].

Since the particle size in the samples was below 300 nm (Figure 4) for the entire period, the decisive process was the increase in the concentration of PVA in the dissolution medium. The colloidal dissolved particles reach the colloid particle size interval and remain in this state during at least 24 h.

Comparing the derived count rates of each carrier dissolved in different media (Table 2) shows that the highest proportion of the PVA walls are being dissolved during the first 120 min in the case of demineralized water, pH = 1.2 solution. Comparing the derived count rates of each carrier dissolved in different media (Table 2) shows that the highest proportion of the PVA walls are being dissolved during the first 120 min in the case of demineralized water, pH = 1.2 solution. However, in the TRIS buffer (pH = 6.8), used in order to avoid the PVA incompatibilities with inorganic phosphate [52], the erosion and formation of colloidal solution process were much slower, and the presence of surface active bile salts did not accelerate the progress. The visible tracking of the solid samples also showed that the intact structure of the solid carriers did not disappeare during the first two hours of the study, but as can be seen in the images and in the weight measurement, at 240 min the carrier is completely disintegrated into colloidal particles (Figure 3, Figure 4 and Figure 5).

### 3.4. Riboflavin Release

Figure 6 shows the release of riboflavin loaded PVA and PLA carriers in pH = 1.2 medium. Carriers without drug delivery orifices were prepared from both biodegradable polymers and dissolution studies were performed. In the case of PLA, which is insoluble in water, no drug release was expected, while in the case of PVA, after a short time a practically linear release of riboflavin was observed. If an orifice for drug release was formed on the support, the drug delivery profile was completely changed, as shown in Figure 6. There were 1, 2, 3, or 4 orifices in the 3D printed carriers. The location and the number of the pores were customized in the CAD design, so the indirect effect of the CAD modifications could have been inspected. The PLA is water-insoluble, so erosion of the body does not affect drug release. This case is clearly controllable by the number of carrier orifices, both the total amount of liberated riboflavin and the rate of drug release. 

The results of the model-dependent evaluation of the dissolution profiles are shown in the third table. The correlations during the fits are between 0.9925 and 0.9999, so they are considered adequate. The kinetic evaluation of the dissolution profiles (Table 3) of riboflavin loaded PLA-based carrier system also show that increasing the number of orifices in the carrier accelerated the dissolution and also increased the maximum amount of drug released, i.e., the infinite value of M increased.

In the instance of the PVA carrier, not only do the orifices play an important role in drug release, but also erosion and deformation of the printed object. This is demonstrated by the fact that the dissolution profile of drug loaded PVA-based systems differs from the PLA-based system. Erosion of PVA is slower in time than the dissolution of riboflavin (Biopharmaceutics Classification System I), so here the PVA carrier can slow drug release. In contrast to the drug loaded PLA carrier, the presence of only one orifice in this system meant almost 100% drug release, since, as our previous studies show, the skeleton was completely dissolved in the first hours of dissolution. As the number of orifices increased, as expected, the *τ_d_* value decreased. It is also important to note that the maximum amount of drug released in the case of PLA printlets is obtained in a shorter time, which is also shown by the low tau values according to the Weibull model; however, due to the rigid, water-insoluble wall, drug occlusion occurs. In the case of PVA carriers, the maximum amount of active ingredient available is higher; there is no such occlusion, but the value of tau is higher compared to PLA, which is probably because the wall material forms a gel not only towards the release medium but also towards the cavity.

## 4. Conclusions

3D printing is going to change the ways of conventional drug manufacturing, and the FDM method enables the researchers to produce high quality tailored dosage forms for each patient according to their individual needs. 

The results demonstrate that dissolution of tested PVA-based 3D printed placebo carriers can be characterized with the appearance of colloidal particles under 300 nm. According to the gravimetry and derived count rate data of the dynamic light scattering method, due to the erosion in the dissolution media the carrier dissolved into colloid state. The most commonly used supporting material can be upgraded into an important excipient filament used in personalized therapy. Due to the characteristics of the polymer, the phenomenon of “ghost tablets” can be abolished because the PVA-based dosage forms biodegrade in the patient’s body in hours. The CAD design ensures the opportunity to add drug-releasing holes into the surface of the carrier. Utilizing this effect, extremely precise individualized treatment can be pharmaceutically engineered, and 3D printed to any patients regarding their needs. With this novel method, standardized drug release can be programmed into the CAD and gcode files of the carriers. The easy modification opportunities ensure extremely wide range of personalized medication. The following experiments would focus on the wider understanding of the behavior of the PVA and other water-soluble filaments.

## Figures and Tables

**Figure 1 materials-14-01350-f001:**
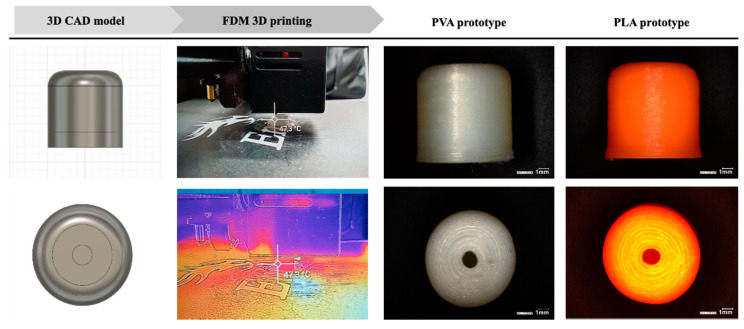
Printlet preparation process and digital microscopic images of polyvinyl alcohol (PVA-) and polylactic acid (PLA)-based prototype with one orifice.

**Figure 2 materials-14-01350-f002:**
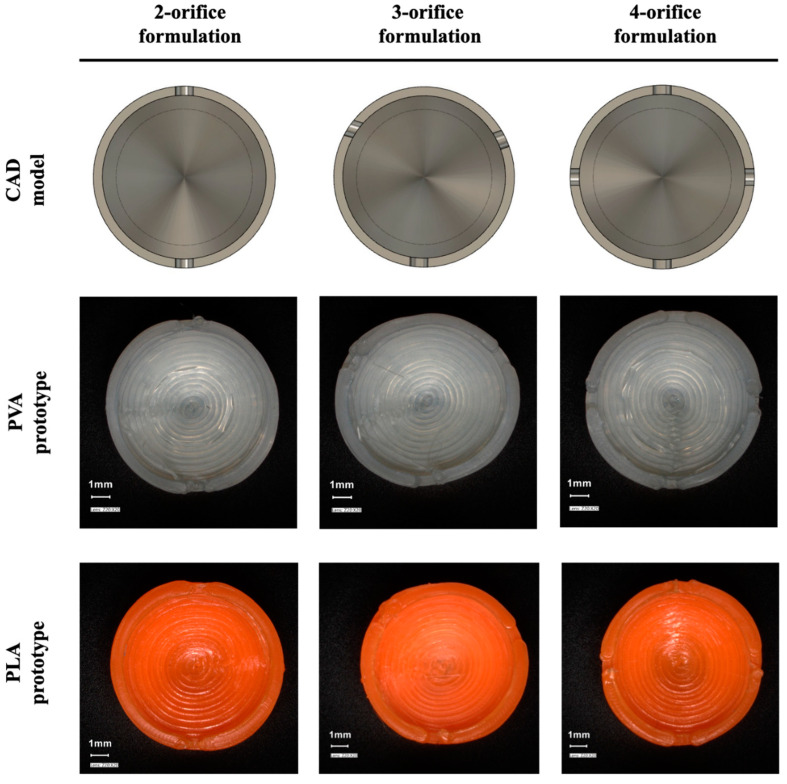
CAD models and digital microscopic images of PVA and PLA-based prototype with multiple orifices.

**Figure 3 materials-14-01350-f003:**
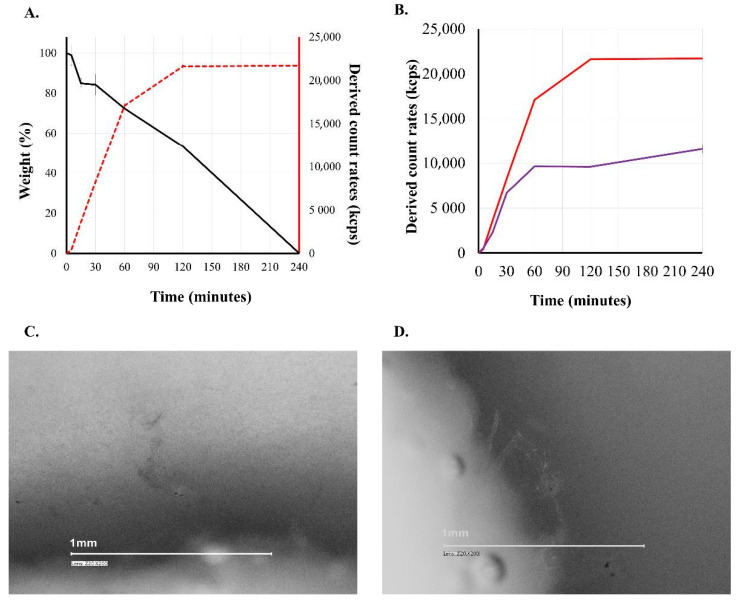
Erosion of PVA-based carrier (**A**): black line—weight loss; red line—pH = 1.2 HCl; (**B**): red line—pH = 1.2 HCl; violet line:—pH = 6.8 phosphate buffer (carrier = PVA; n = 3; mean ± SD). (**C**,**D**): digital images of PVA wall during dissolution (pH = 1.2; time = 90 min).

**Figure 4 materials-14-01350-f004:**
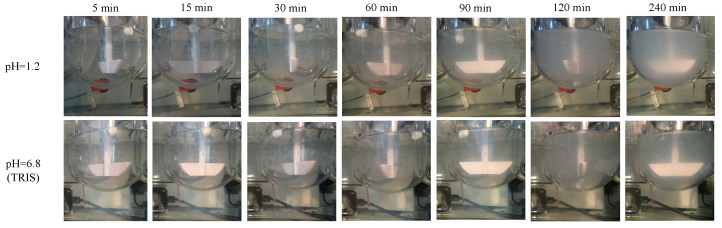
Visual tracking of the PVA-based carrier erosion in different media.

**Figure 5 materials-14-01350-f005:**
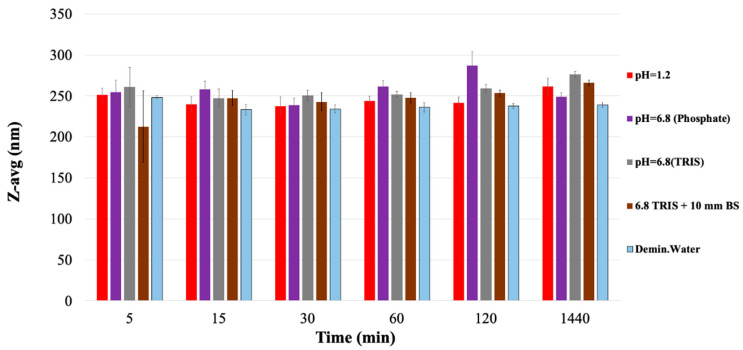
Particle size of dissolution sample (carrier = PVA; n = 3; mean ± SD).

**Figure 6 materials-14-01350-f006:**
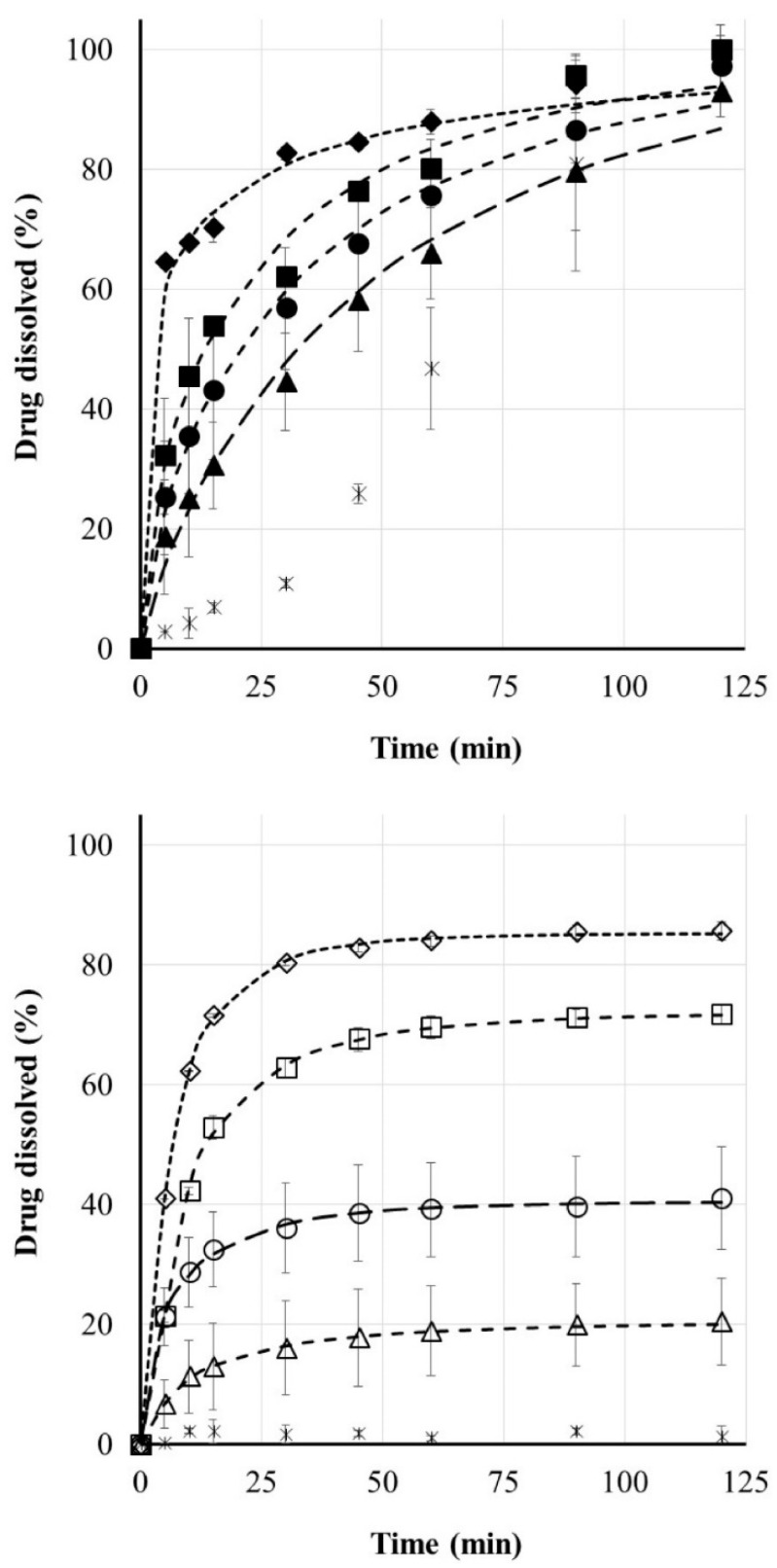
Drug release profile of riboflavin loaded PLA and PVA carriers printed containing various number of orifices (zero—cross; one—triangle; two—circle; three—square; four—diamond; n = 3; mean ± SD) with fitted Weibull (dotted lines).

**Table 1 materials-14-01350-t001:** Physical characteristics of 3D printed carriers.

	PVA Carrier	PLA Carrier
Weight (g) (n = 20; mean ± SD)	0.42 ± 0.007	0.49 ± 0.004
Height (mm) (n = 20; mean ± SD)	9.75 ± 0.053	9.75 ± 0.036
Diameter (mm) (n = 20; mean ± SD)	9.66 ± 0.298	9.66 ± 0.239
Hardness (N) (n = 10; mean ± SD)	212.63 ± 75.87	300.0 ± 1.00
Friability (%)	0.016	0.024

**Table 2 materials-14-01350-t002:** The percentages of derived count rates *(DCR_t_ /DCR_1440min_* × *100)* of various dissolution media (carrier = PVA; n = 3; mean ± SD).

Medium	5 min	15 min	30 min	60 min	120 min	1440 min
pH = 1.2	1.83 ± 0.92	15.94 ± 1.26	36.3 ± 1.4	75.15 ± 0.92	94.79 ± 0.66	100 ± 1.18
pH = 6.8 (Phosphate)	4.49 ± 0.46	19.77 ± 0.38	57.83 ± 1.28	83.21 ± 0.42	85.76 ± 1.01	100 ± 1.49
pH = 6.8 (TRIS)	0.55 ± 0.15	2.81 ± 0.4	10.95 ± 2.61	19.33 ± 0.11	36.0 ± 0.43	100 ± 1.01
pH = 6.8 (TRIS) + 10 mm BS	1.0 ± 0.1	3.57 ± 1.56	7.4 ± 1.02	13.8 ± 1.12	25.7 ± 1.54	100 ± 6.9
Demineralized water	2.37 ± 1.0	10.97 ± 0.76	30.01 ± 0.61	50.76 ± 0.8	73.89 ± 0.52	100 ± 0.54

**Table 3 materials-14-01350-t003:** Kinetic parameters of dissolution estimated according to Weibull distribution function (pH = 1.2).

Filament Base	PLA				PVA			
Number of orifices	1	2	3	4	1	2	3	4
*M_∞_* (%)	20.46	40.52	72.15	85.28	100.00	100.00	99.92	100.00
*t_0_* (min)	2.42	0.11	3.71	2.43	0.00	0.00	0.00	0.00
*τ_d_* (min)	12.17	7.54	7.53	5.00	50.77	34.48	23.96	6.90
*β*	0.59	0.62	0.60	0.63	0.82	0.70	0.64	0.34
*r*	0.9991	0.9992	0.9999	0.9999	0.9947	0.9960	0.9925	0.9950

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
