# Peer review of "Evaluation of Biodegradable PVA-Based 3D Printed Carriers during Dissolution"

_materials, 2021, doi:10.3390/ma14061350_

Round 1

Reviewer 1 Report

The paper titled Evaluation of biodegradable PVA based 3D printed carriers during dissolution concerns an important and present problem regarding new solutions that can be used in targeted or multi-drug therapies. The presented concept seems to be interesting scientific solution, however the prepared manuscript lack of some information and what is more important proper result discussion.

In line 115 is not enough information (polymer characterization) about PVA and PCL.

In lines 138-144 printing parameters are missing (for example the temperatures).

In lines 155-156 there no parameters which were set for the conducted tests?

In line 239 there is no specific data confirming the wording ‘very low friability’

Beside that the authors did not comment/explained (in general did not discussed):

- why the PVA carrier has such big hardness standard deviation,

- from where does the differences in the percentages of derived count rates various dissolution media come ?

- how these differences can influence the usability and so on of the carrier

- why in case of PLA carrier, the carriers having one, two or three orifices did not reached higher value of drug dissolution (Figure 5A)?

Author Response

Dear Reviewer,

Thank You for your time.
The responses for the comments and the corrections are attached in the pdf file.

Reviewer 2 Report

Manuscript ID: materials-1110511
Title: Evaluation of biodegradable PVA based 3D printed carriers during dissolution

Authors: Bálint Basa , Géza Jakab , Nikolett Kállai-Szabó , Bence Borbás , Viktor Fülöp , Emese Balogh , István Antal

Authors described two 3D printed polymeric carriers for model riboflavin drug release. The erosion of PVA carrier in different pH water solutions was evaluated. While authors declare the effect of ionic strength on dissolution characteristics, no ions concentration dependency did not appear in the results. Only different buffers were evaluated. Manuscript is well written and describes actual methodology of targeting drug delivery. Conclusions are supported by the results obtained in well-established experiments.

Comments:

It is not clear from the text, what are particles measured by DLS formed during dissolution of PVA carriers. Also, more discussion why they formed constantly ~ 300 nm in size in particular time of dissolution could contribute to better understanding.

The readers would appreciate more discussion about the differences of drug release from PLA and PVA carriers. There is no scheme how 2,3 and 4 orifices carrier look like. The release of drug is also much affected by stirring/shaking of carrier during release test. The specification is missing in the experimental part. This could be also connected with the fact, that 1 orifice carrier did not release drug completely with expected longer time.

What is the reason of slowing down the drug release from PVA carrier? It could be also interesting to compare PVA carrier without the orifice. Due to the dissolution of PVA, the drug release could be linearized and should be proportional to PVA dissolution.

In conclusion, I do recommend to accept this article for Materials journal after revision. 

Author Response

(The authors gave the same response as above.)

Reviewer 3 Report

There are some weaknesses through the manuscript which need improvement. Therefore, the submitted manuscript cannot be accepted for publication in this form, but it has a chance of acceptance after a major revision. My comments and suggestions are as follows:

1- Abstract gives information on the main feature of the performed study, but some details about the examination methods must be added.

2- Authors must clarify necessity of the performed research. Objectives of the study, and also differences with the previous researches must be clearly mentioned in introduction.

3- The literature study must be enriched. In this respect, authors must read and refer to the following papers: (a) buckling in thin walled members: https://doi.org/10.1016/j.tws.2019.01.041(b) environmental impact of additive manufacturing: https://doi.org/10.1016/j.apmt.2020.100689 and (c) biodegradable polymer: https://doi.org/10.1016/j.biortech.2021.124739

4- As it is an experimental investigations, authors must add some figures (real) to show test coupons, and specimens under test conditions.

5- Authors must clarify why these particular materials were selected and tested.

6- The printing process must be explained in details (text and figures must be added to 2.2).

7- In 2.5.1 it was mentioned that “The determination of erosion of PVA samples was examined by weight loss”. This issue must be explained in detail, and data (weigh loss values) must be presented.

8- Figures must be illustrated in high quality (e.g., quality of Fig. 3 is too low). Font size in all figures must be double checked (e.g., font size in Fig. 4 is large).

9- The main reference of each formula must be cited. Moreover, each parameters in equations must be introduced. Please double check this issue.

10- In its language layer, the manuscript should be considered for English language editing. There are sentences which have to be rewritten.

11- The conclusion must be more than just a summary of the manuscript. List of references must be updated based on the proposed papers. Please provide all changes by red color in the revised version.

Author Response

(The authors gave the same response as above.)

Round 2

Reviewer 1 Report

Thank the Authors for answering to my all comments and concerns. The effort taken has improved the manuscript significantly.

I would just suggest the author to check if in Materials and Methods should not be a 'density' instead of thickness (lines 133 and 134).

Author Response

Dear Reviewer,

Thank you so much for the positive feedback.

Reviewer 2 Report

Manuscript ID: materials-1110511 - revision
Title: Evaluation of biodegradable PVA based 3D printed carriers during dissolution

Authors: Bálint Basa , Géza Jakab , Nikolett Kállai-Szabó , Bence Borbás , Viktor Fülöp , Emese Balogh , István Antal

There is a lot of typo mistakes in the revised version, mainly in the added text.

Comments:

It is not clear from the text, what are particles measured by DLS formed during dissolution of PVA carriers. Also, more discussion why they formed constantly ~ 300 nm in size in particular time of dissolution could contribute to better understanding. What particles are formed from “water soluble” PVA? This was my first question/note in previous review, however I am not satisfy with presented answer.

Other comments were completed

In conclusion, I do recommend to accept this article for Materials journal after revision.

Author Response

Dear Reviewer,

Thank You for the important comment. We tried to answer it better. In order to achieve this, new reference, figure and modifications in the text were supplemented.

Reviewer 3 Report

In the submitted revised manuscript, most of the reviewers' comments have been properly responded and corresponding modifications have been conducted. I think it can be considered for publication.

Author Response

Dear Reviewer,

Thank you for the positive feedback.
